# FEED: Feature-level Ensemble Effect for knowledge Distillation

## Abstract

This paper proposes a versatile and powerful training algorithm named *Feature-level Ensemble Effect for knowledge Distillation* (FEED), which is inspired by the work of *factor transfer*. The *factor transfer* is one of the knowledge transfer methods that improves the performance of a student network with a strong teacher network. It transfers the knowledge of a teacher in the feature map level using high-capacity teacher network, and our training algorithm FEED is an extension of it. FEED aims to transfer ensemble knowledge, using either multiple teacher in parallel or multiple training sequences. Adapting peer-teaching framework, we introduce a couple of training algorithms that transfer ensemble knowledge to the student at the feature map level, both of which help the student network find more generalized solutions in the parameter space. Experimental results on CIFAR-100 and ImageNet show that our method, FEED, has clear performance enhancements, without introducing any additional parameters or computations at test time.

## 1 Introduction

Recent successes of CNNs have led to the use of deep learning in real-world applications. In order to manipulate these deep learning models, people are asking deep CNNs to use multi-class datasets to find manifolds separating different classes. To meet this need, deep and parameter-rich networks have emerged that have the power to find manifolds for large numbers of classes. However, these deep CNNs suffer from the problem of overfitting due to their great depth and complexity, which results in the drop of performance at the test time. In fact, even a small ResNet applied for a dataset such as CIFAR-100 (Krizhevsky et al.) will not have room to learn more because the train losses converge, whereas the test accuracy is significantly lower. This phenomena have led to the need of learning DNN models with appropriate regularization to allow them to generalize better. In fact, regularizing a model to achieve high performance for new inputs is a technique that has been used since the era of early machine learning.

Among them, model ensemble (Dietterich, 2000) is one of the popular regularization methods (Goodfellow et al., 2016), which has been used as a way of alleviating the problem of overfitting in a single model. But it has drawbacks in that it requires multiple models and inputs should be fed to each of them at test time. For a solution of this problem, Hinton et al. (2015) proposed *Knowledge Distillation* (KD) which trains a student network using soft labels from ensemble models or a high-capacity model. They obtained meaningful results in speech recognition dataset and this work brought advent of *knowledge transfer*, an area in the representation learning that aims performance improvements by training a weak student network by giving various forms of knowledge of expert teacher networks (Huang & Wang, 2017). It is also categorized as one family of model compression (Kim et al., 2018), since it helps the student network achieve higher accuracy with fixed number of parameters given.

The recent knowledge transfer algorithms can be approximately categorized in two ways. The first way is whether to use an ensemble model as a teacher or a single high-capacity model as a teacher. The second is whether to transfer the teacher's prediction or the information from feature maps. Whereas the methods that use teacher's prediction can use both types of teachers, to the best of our knowledge, methods using feature-map-level information only use a single high-capacity model as a teacher. For example, studies of *Factor Transfer* (FT) (Kim et al., 2018), *Attention Transfer* (AT) (Zagoruyko & Komodakis, 2016a), and *Neuron Selectivity Transfer* (NST) (Huang & Wang, 2017) set

the student network as a shallow network with a small-sized parameters, and set the teacher network as a deeper and more powerful network instead of an ensemble expert. One of the drawbacks of methods with a high-capacity teacher model is that high-capacity model may be hard to obtain (Lan et al., 2018).

On the contrary, the ones which use an ensemble of networks can transfer ensemble knowledge and also have an advantage of peer-teaching framework (Hinton et al., 2015; Lan et al., 2018; Furlanello et al., 2018). Also, Zhang et al. (2017) showed that using one same type of network for transferring output-level knowledge can improve performance of a network. However, the methods that delivers knowledge at the feature-map level also has advantages that they can give more specific information to the student compared to the methods that only rely on the output predictions of the teacher.

To make full advantages of both ensemble teacher method and feature map methods, we came up with a new framework that delivers knowledge of multiple networks at the feature-map level. In this paper, we train a new student network using the same type of teacher networks using the scheme of FT, a decent knowledge transfer algorithm that transfers knowledge using auto-encoding *factors*. We utilized the *translator* of FT to transfer ensemble knowledge, introducing two different kinds of ensemble-effective training algorithms:

- We recursively set the trained student network as a new teacher network which helps training a new student, accumulating knowledge in ensemble at the feature map level.

- We transfer knowledge to the student from multiple teachers simultaneously, expecting the student network to learn ensemble knowledge at the feature map level.

The paper is organized as follows. First, we briefly explain the related works including the method FT. Then a couple of more advanced versions of FT are proposed. Next, we verify our proposed methods with experiments. Experimental results from our proposed training methods are compared with the case of KD on CIFAR-100 and ImageNet datasets. Finally, we compare our method with other kinds of recent knowledge transfer algorithms.

## 2 RELATED WORKS

Many researchers studied the ways to train models other than using a purely supervised loss. In early times of these studies, *Model Compression* (Bucila et al., 2006) studied the ways to compress information from ensemble models in one network. More recently, Hinton et al. (2015) proposed Knowledge Distillation (KD), which uses softened softmax labels from teacher networks when training the student network, and motivated many researchers to develop many variants of it to various domains. The paper has been applied to transfer learning, and also led the advent of knowledge transfer with applications to many domains.

FitNet (Romero et al., 2014) applied KD to train a student network with different capacity to find better trade off between its accuracy and time cost. Attention Transfer (Zagoruyko & Komodakis, 2016a) tried to transfer the attention map of the teacher network to the student network, and got meaningful results in knowledge transfer and transfer learning tasks. Huang & Wang (2017) also tried to match the feature of the student network and teacher network devising a loss term MMD (Maximum Mean Discrepancy). Yim et al. (2017) introduced another knowledge transfer technique for faster optimization and applied it also for transfer learning.

Differently from previous works, Abdulnabi et al. (2018) and Tang et al. (2016) tried to transfer information in RNNs. Sharing knowledges were applied to even NAS (Neural Architecture Search) (Zoph & Le, 2016), making ENAS (Efficient NAS) (Pham et al., 2018) to overcome the heavy time cost. Tarvainen & Valpola (2017) used Mean Teachers to train a student network by a semi-supervised manner, and Radosavovic et al. (2017) expanded the semi-supervised learning to omni-supervised learning by proposing Data Distillation, a more applicable version of knowledge distillation. Recently, Transparent Model Distillation (Tan et al., 2018) tried to investigate the model distillation for transparency.

# 3 FACTOR TRANSFER

FT (Kim et al., 2018) is one of knowledge transfer methods that uses a pair of paraphraser and translator as a mediator of the teacher and the student networks. The *paraphraser* and the *translator* are attached to the last convolution layer of the teacher and the student networks, respectively and the translator is trained in a way that its output assimilates the output of the paraphraser.

## 3.1 TEACHER FACTOR EXTRACTION WITH PARAPHRASER

The paraphaser is trained in an unsupervised manner to map the teacher's knowledge into another form so that the student network understands the knowledge more easily. It is designed with several convolution layers and transposed-convolution layers, in a similar way to autoencoders. The teacher's paraphrased knowledge, called teacher factor ($F_T$), is output of the last convolution layer in the paraphraser. The paraphraser is trained with simple reconstruction loss function,

$$\mathcal{L}_{rec} = \|x - P(x)\|^2, \tag{1}$$

where the paraphraser $P(\cdot)$ takes the featuremap $x$ as an input. The convolution and the transposed-convolution layers in the prarphraser are designed to make the spatial dimension of the teacher factor the same as that of the input, but the depth (number of channels) of the teacher factor can be made different with that of the input with the a rate of $k$.

## 3.2 FACTOR TRANSFER WITH TRANSLATOR

The translator is designed with several convolution layers, whose output is called student factor ($F_S$). The student network and the translator are trained simultaneously with the the combined loss consisting of the conventional cross entropy loss $\mathcal{L}_{cls}$ and the factor transfer loss $\mathcal{L}_{FT}$ as follows:

$$\mathcal{L}_{student} = \mathcal{L}_{cls} + \beta \mathcal{L}_{FT}, \tag{2}$$

$$\mathcal{L}_{FT} = \left\| \frac{F_T}{\|F_T\|_2} - \frac{F_S}{\|F_S\|_2} \right\|_p^p, \tag{3}$$

where $\|A\|_p$ is the $p$-norm of the vectorized version of a tensor $A$.

We train the translator to output $F_S$ that mimics $F_T$. The training is done in an end-to-end manner so that the gradients from the translator also affect the student network. The translator is used only in the training phase, which means that the factor transfer method does not affect the computation at test time.

# 4 PROPOSED TRAINING ALGORITHMS

In deep CNNs, due mainly to the curse of dimensionality, the data points that lie on the data space are very sparse, and this phenomena can be easily detected by applying algorithms like $k$-nearest neighbors. Necessarily, decision boundaries that determine the borders dividing classes are multitudinous, because finding boundaries that fit well to a training dataset is relatively an easy task. Even if the networks with the same architecture are trained, the learned decision boundaries cannot be the same. This is why ensemble methods usually perform better than a single model despite of their structural equality. Goodfellow et al. (2016) also state that different models will not make all the same errors on the test dataset.

Consider the conditions that determine the training procedure of CNNs. They include the structure of the CNN and the choice of an optimizer, random initialization, the sequence of mini-batch, and the types of data augmentations. If you make the same conditions for two different CNNs, their training procedure will be identical. However, we usually determine only the structure of the CNN, usually keeping the others to be random. Consequently, two networks with the same structure will definitely not learn the same decision boundaries.

Additionally, Kim et al. (2018) stated that they resolve the 'inherent difference' between two networks. Among the inherent differences, minimizing the differences in the structure of CNNs can help better

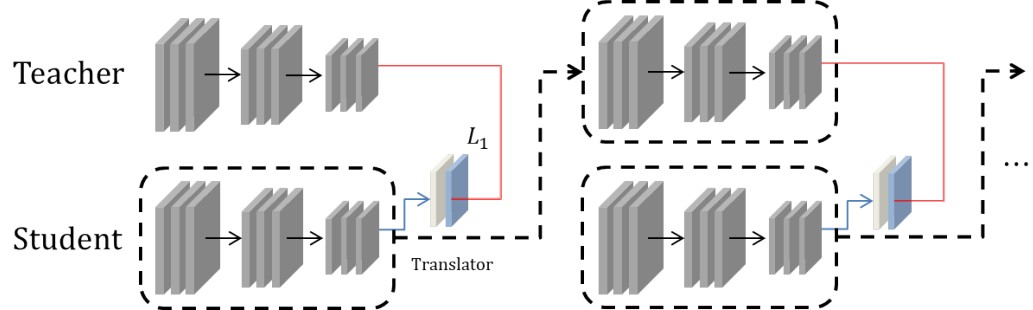

Figure 1: The sequential FEED. The first FT stage is the same as the original work in (Kim et al., 2018), except that the teacher network and the student network have the same architecture. The dotted line means that the trained student network is used as a teacher network for the next stage. This training procedure is repeatedly performed over several stages to deliver ensemble knowledge to the successive student.

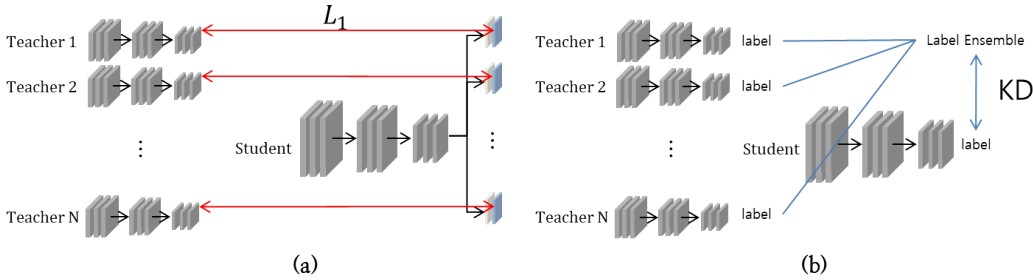

Figure 2: Illustration of our second proposed algorithm and KD to compare the differences. (a) Full view of our second proposed training procedure, the parallel FEED. (b) Full view of the training procedure of KD

learn the knowledge of the teacher network. This motivation provides us chances to propose several modified versions of existing methods. In this section, we explain the two feature-level ensemble training algorithms that we use for boosting the performance of a student network without introducing any additional calculations at test time. The proposed methods are collectively called as FEED which is an abbreviation for the *Feature-level Ensemble Effect for knowledge Distillation*.

## 4.1 SEQUENTIAL FEED

Taking the advantage that the structure of the teacher network and the student network are the same, we propose a learning method that can accumulate and assemble knowledge by performing knowledge transfer several times. We name this algorithm as *sFEED (sequential FEED)*, and the training procedure is illustrated in Figure 1. The paraphraser is omitted for sFEED, because the *inherent difference* which FT mentioned on their paper is extenuated by choosing the same type of teacher network and the student network. Additionally, omitting the paraphraser simplifies the training procedure. Omitting it not only eliminates the ambiguity of the choices of hyper-parameters, but also reduces the three-stage training procedure of FT into two stages.

Since the teacher and the student networks share the same architecture, problems due from different types of architectures being used for the teacher and the student do not occur here. A similar approach has been introduced in the paper of Furlanello et al. (2018), using output predictions. If the student network is trained standalone, it would perform similar to the teacher network. But from the view of knowledge ensemble, since the teacher network delivers feature-level knowledge different from that of the student network, the student network will benefit from it.

### 4.2 PARALLEL FEED

Assuming that giving knowledge in feature map level and giving knowledge in ensemble way both have their advantages, we wanted to make cooperation of these two kind of methods. Tackling this problem, we propose a training structure named *pFEED (parallel FEED)* to transfer the ensemble knowledge at the feature map, and is our second main proposed method. In this way, we take advantage of both the label-based method of delivering the information of the ensemble model and the feature-based method of delivering more specific information than the label-based method. The proposed algorithm is compared with KD which distills the knowledge of teacher network with ensemble label. Our algorithm is shown in Figure2.(a), and the training method of KD is shown in Figure2.(b). Consistent with what sFEED did, we omitted the paraphraser also in pFEED. Unlike the original FT, we use multiple teachers with multiple translators. If we use k different teachers, the loss term will be like following.

$$\mathcal{L}_{student} = \mathcal{L}_{cls} + \beta \sum_{n=1}^{k} \mathcal{L}_{FT_n}, \tag{4}$$

$$\mathcal{L}_{FT_n} = \left\| \frac{x_n}{\|x_n\|_2} - \frac{F_S}{\|F_S\|_2} \right\|_1. \tag{5}$$

$\mathcal{L}_{FT_n}$ is the FT loss from $n^{th}$ teacher network, and $x_n$ is the output feature map obtained from $n^{th}$ teacher network.

## 5 EXPERIMENTS

In this section, we first show the classification results on CIFAR-100 (Krizhevsky et al.). Second, we explore the feasibility of our algorithm on Imagenet (Russakovsky et al., 2015), a commonly used large dataset. For both datasets, we apply our algorithm and compare the results with KD, and analyze the results quantitatively. Third, we compare our results the state-of-the-art model on CIFAR-100. In the remaining section, we show some analysis on our algorithm and explain the experimental details.

We chose 3 types of CNNs to check the applicability of our algorithms on CIFAR-100: ResNet (He et al., 2016), Wide ResNet (Zagoruyko & Komodakis, 2016b), and RexNext (Xie et al., 2017). For ResNets, we chose ResNet-56 and ResNet-110 which have fewer number of parameters compared to recent CNNs, and WRN28-10 is a model that controls the widen factor, with much more number of parameters. WRN28-10 model achieves the best classification accuracy on CIFAR-100 among the WRNs reported on their parer. The ResNext29-16x64d also achieves the best classification accuracy on CIFAR-100 in their parer, and this type of CNNs controls the cardinality of CNNs and it has much more parameters compared to other models. For ImageNet, we used ResNet-34 to confirm the feasibility on large scale datasets.

### 5.1 SEQUENTIAL FEED

The classification results of our algorithm on CIFAR-100 can be found on Table 1. The word 'Stack' on Table 1 is the number of recursions that the student model is trained. Table 2 shows the results using the paraphraser. Although the existence of the paraphraser seems to affect the performance, omitting it seemed better for stronger networks. Additionally, as mentioned earlier, omitting the paraphraser simplifies the training procedure.

In many cases, the classification accuracy improves as the number of stacks increases. Though they have some fluctuations, it might be possible to achieve higher accuracy. We only experimented up to 5 times since all of them achieves fairly good enough accuracy compared to baseline models. Note that even though we train translators on the student side, they are not counted in Params column because it is not used at the test phase.

The results of sFEED for ImageNet is on Table 3. For the base model, we simply used the pre-trained model that Pytorch supplies, and could achieve a desired result that the performance of Top-1 and Top-5 accuracy improves at each stack.

| Model Type | Scratch* | Scratch | Stack2 | Stack3 | Stack4 | Stack5 | Params |
|---|---|---|---|---|---|---|---|
| ResNet-56 | – | 28.03 | 26.02 | 26.00 | 25.59 | **25.33** | 0.85M |
| ResNet-110 | – | 27.14 | 25.25 | **24.33** | 24.58 | 24.40 | 1.73M |
| WRN28-10 | 19.25 | 18.96 | 17.68 | 17.50 | 17.52 | **17.27** | 36.5M |
| ResNext29-16x64d | 17.31 | 17.41 | 16.80 | 16.47 | 16.22 | **15.94** | 68.1M |

Table 1: Test classification error (%) of sFEED on CIFAR-100 dataset. The model's scores on the **Scratch\*** column are the same as the scores reported on their original papers, and those on the **Scratch** column are from our implementation. The parameters are counted in Millions.

| Model Type | Scratch* | Scratch | Stack2 | Stack3 | Stack4 | Stack5 | Params |
|---|---|---|---|---|---|---|---|
| ResNet-56 | – | 28.23 | 26.33 | 25.70 | 25.91 | **25.18** | 0.85M |
| ResNet-110 | – | 26.91 | 24.90 | 24.50 | 24.34 | **24.13** | 1.73M |
| WRN28-10 | 19.25 | 19.00 | 18.23 | 18.05 | 18.14 | **17.84** | 36.5M |
| ResNext29-16x64d | 17.31 | 17.41 | 16.80 | 16.76 | **16.47** | 16.48 | 68.1M |

Table 2: Test classification error (%) for sFEED on CIFAR-100 using the paraphraser.

## 5.2 PARALLEL FEED

In experiments on pFEED, we used the same type of networks that were used on previous experiments. For all 4 types of CNNs, we compared the classification results with the result of KD because we designed our training algorithm with intention of receiving more ensemble-like knowledge from multiple teachers. The difference between pFEED and KD is that KD ensembles the knowledge on the output level, and our proposed method try to draw the effect of ensemble on feature-map level, for delivering more specific information. The results are on Table 4.

The 'Scratch' column shows the performance of the base networks, used in KD for model ensemble, and also used as teachers in pFEED. For all experiments, pFEED consistently got higher accuracy compared with KD. It is worth noting that the performance of KD is almost equivalent to pFEED for small networks, but as it comes to the networks with larger number of parameters, pFEED shows better accuracy compared to KD. This result matches the hypothesis that out delivering feature-map-level information will provide more specific information to the student.

The results of pFEED for ImageNet is on Table 5. We also could find some accuracy improvements on ImageNet dataset, but did not have enough resources to train models with larger parameters, and used only three teachers. We could get decent results, but improvements are not strong as those of sFEED on ImageNet.

## 5.3 QUALITATIVE ANALYSIS

**Reconstruction Loss:** A paraphraser in the FT can be interpreted as a convolutional autoencoder in that it uses convolution and transposed convolution layers with a reconstruction loss, then the factor can be interpreted as a latent vector $z$. Supposing that the reason for the accuracy gains shown in the previous tables is that the student learns the ensemble knowledge, student network is forced to learn information with high complexity. Let us denote the input of paraphraser as $x$. The increase in the complexity of feature representation is equivalent to the increase in the complexity of $x$. In

| Model Type | Scratch* | Stack2 | Stack3 | Stack4 | Stack5 |
|---|---|---|---|---|---|
| ResNet-34(Top-1) | 26.45 | 25.60 | 25.30 | 25.18 | **25.00** |
| ResNet-34(Top-5) | 8.54 | 8.08 | 7.86 | **7.73** | 7.83 |

Table 3: Validation classification error (%) of sFEED on Imagenet dataset. The model's scores on the **Scratch\*** column are the same as the scores reported on the Pytorch implementation.

| Model Type | Scratch (5 runs) | | | | | KD | P-FEED | Ens |
|---|---|---|---|---|---|---|---|---|
| ResNet-56 | 27.98 | 28.18 | 28.48 | 28.05 | 28.21 | 24.69 | **24.64** | 22.45 |
| ResNet-110 | 27.14 | 26.70 | 26.86 | 27.13 | 27.01 | 23.50 | **22.84** | 21.20 |
| WRN28-10 | 18.94 | 18.94 | 19.18 | 19.24 | 19.17 | 18.30 | **17.10** | 16.59 |
| ResNext29-16x64d | 17.27 | 17.30 | 17.40 | 17.43 | 17.20 | 16.64 | **15.62** | 15.66 |

Table 4: Test classification error (%) of pFEED on CIFAR-100 dataset. The numbers in Scratch column are from our implementation. The KD column is our reproduction of KD, and the Ens column is the ensemble score of 5 models, each of which uses purely cross-entropy loss.

| Model Type | Scratch (5 runs) | | | | | pFEED |
|---|---|---|---|---|---|---|
| ResNet-34(Top-1) | 26.45 | 26.59 | 26.40 | 26.77 | 26.64 | **25.27** |
| ResNet-34(Top-5) | 8.54 | 8.72 | 8.63 | 8.68 | 8.61 | **7.79** |

Table 5: Validation classification error (%) of pFEED on Imagenet dataset.

FEED training, since the number of parameters in the paraphraser is fixed, the size of $z$ also should be fixed. Consequently with complexity of $x$ increasing, $p(x|z)$ decreases, resulting in the increase of the reconstruction loss.

For both of our proposed training methods, we recorded the average training reconstruction losses of the paraphrasers normalized by the size of the paraphraser and plotted the curve on Figure 3 and Figure 4 (though we do not actually use paraphrasers for FEED training). In Fig. 3, Ph1 through Ph4 are the paraphrasers trained based on the student networks of Stack1 through Stack4 in Table 1, and the paraphrasers in Figure 4 is trained based on one of the scratch teacher networks and following student network of pFEED in Table 4. As expected, as the knowledge is transferred, the reconstruction loss becomes larger which indicates that the student network learns more difficult knowledge and thus the classifier accuracy increases. This trend surprisingly matches the results on the tables. In Figure 3, the legends matches the trends of ResNet-56 row in Table 1 of sFEED (especially, Stack 2 and 3 have similar errors and likewise, Ph2 and Ph3 are similar), and the big legend in Figure 4 also follows the high performance increase in WRN28-10 row of Table 4.

## 5.4 IMPLEMENTATION DETAILS

**CIFAR-100**: In the student network training phase, we used $l_1(p = 1)$ loss and the hyper-parameter $\beta$ in eq.2 was set to 500 for ResNets and 2,000 for WideResNet and ResNext. We tried to set the training procedure to be the same as that of the original paper. For ResNets, we set the initial learning rate to 0.1 and decayed the learning rate with rate of 0.1 at 80, 120 epochs, and training finished at 160 epochs. For WideResNets, we set the initial learning rate to 0.1 and decayed the learning rate with rate of 0.2 at 60, 120, 160 epochs, and ended the training at 200 epochs. For ResNexts, we set the initial learning rate to 0.1 and decayed the learning rate with rate of 0.1 at 150, 225 epochs, and training finished at 300 epochs. For all the experiments, simple SGD is used as an optimizer, with momentum of 0.9 and weight decay of $5 \times 10^{-4}$, and mini-batch size of 128. The ResNets and WideResNets were trained on single Titap XP and the ResNexts were trained on four 1080 ti GPUs. The same setting was applied for translators with 3 convolution layers, since the translators were trained jointly with the student network.

**Paraphraser:** All the paraphrasers used in Table 2 are trained for 10 epochs, with learning rate of 0.1. The original paper states that the paraphrasers are trained for 30 epochs, but training it over 10 epochs was unnecessary. All paraphrasers has 3 convolution layers, and since the teacher network and student networks are the same kind, we set paraphrase rate $k$ to be 1 for simple implementation. According to the paper of FT, the choice of it did not seem to affect the performance much.

**ImageNet**: The hyper-parameter $\beta$ was set to 1,000, and following the training schedule of the Pytorch framework, train starts with learning rate of 0.1 and decays by the factor of 0.1 at 30, 60 epochs and finishes at 90 epoch, with mini-batch size of 256. All other conditions are set to be the same as the setting of CIFAR-100.

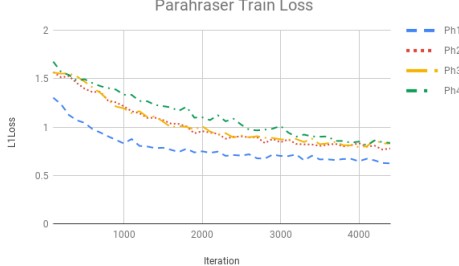
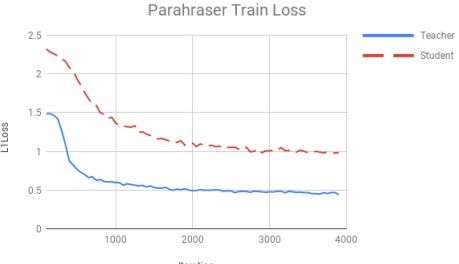

Figure 3: Paraphraser reconstruction loss $\mathcal{L}_{rec}$(training) for Resnet-56 with sFEED.

Figure 4: Paraphraser reconstruction loss $\mathcal{L}_{rec}$(training) for WRN28-10 with pFEED.

|  | Algorithm | Model Type | Err. | Params |
|---|---|---|---|---|
| CIFAR-100 | ONE | ResNet-110 | 21.62 | 1.7M |
|  |  | ResNext29-8x64d* | 16.07 | 34.4M |
|  | BAN | WRN28-10(BAN-1) | 18.25 | 36.5M |
|  |  | DenseNet-80-80(BAN-3) | **15.50** | 22.4M |
|  | FEED | ResNet-110(pFEED) | 22.84 | 1.7M |
|  |  | WRN28-10(sFEED) | 17.27 | 36.5M |
|  |  | WRN28-10(pFEED) | 17.10 | 36.5M |
|  |  | ResNext29-16x64d(sFEED) | 15.94 | 68.1M |
|  |  | ResNext29-16x64d(pFEED) | **15.62** | 68.1M |

Table 6: Test classification error (%) of the models on CIFAR-100 dataset. The errors listed for each methods are the numbers from their original papers, without reproducing on our own. The parameters are counted in Millions.

**Setting of** $\beta$: For the ease of re-producing, the choice of $\beta$ is important. Supposing that we use $\mathcal{L}_{FT}$ with $l_1(p = 1)$, the the loss scale of $\mathcal{L}_{FT}$ depends on the number of nodes. Empirically, if one of the scales of either $\mathcal{L}_{cls}$ or $\mathcal{L}_{FT}$ is dominant, the accuracy diminishes compared to the even case, so we approximately adjusted the scale of $\mathcal{L}_{FT}$, resulting in different $\beta$s in different networks.

**Hyperparameters of KD**: We set the Temperature $T$ for softened softmax to 4 as in KD (Hinton et al., 2015)

## 5.5 COMPARISON WITH KNOWLEDGE TRANSFER METHODS

The comparison with recent state-of-the-art knowledge distillation methods are on Table 6. The scores on ONE methods are from the paper of Lan et al. (2018), and the scores on the BAN methods are from the paper of Furlanello et al. (2018). Both methods use final predictions to transfer knowledge, while our method uses output feature maps. Unfortunately, we could not prepare all variations of networks and hyper-parameters for fair comparison, but could achieve decent performances.

## 6 CONCLUSION

In this work, we proposed a couple of new network training algorithms referred to as *Feature-level Ensemble Effect for knowledge Distillation* (FEED). With FEED, we can improve the performance of a network by trying to inject ensemble knowledge to the student network. The first one, sequential FEED recursively trains the student network and incrementally improves performance. The second one, parallel FEED trains the student network using multiple teachers simultaneously. The qualitative analysis with reconstruction loss gives hints about the cause of accuracy gains. The main drawback is the training times needed for multiple teachers which is an inherent characteristics of any ensemble methods, and pFEED causes bottleneck by feeding inputs to multiple teachers. Consequently, applying it more efficiently will be our future work, together with application to other domains.

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
