# OpenReview forum: "FEED: Feature-level Ensemble Effect for knowledge Distillation"
_ICLR.cc/2019/Conference_

### Official Review · AnonReviewer1 · 2018-10-30
**A review**

**Rating:** 4
**Confidence:** 4

**Review:**

In this paper, the authors present two methods, Sequential and Parallel-FEED for learning student networks that share architectures with their teacher.

Firstly, it would be a good idea to cite https://arxiv.org/abs/1312.6184, it precedes knowledge distillation and is basically the same thing minus a temperature parameter and a catchy name.

The paper could do with some further grammar/spell checks.

It isn't clear to me where the novelty lies in this work. Sequential-FEED appears to be identical to BANs (https://arxiv.org/abs/1805.04770) with an additional non-linear transformation on the network outputs as in https://arxiv.org/abs/1802.04977. Parallel-FEED is just an ensemble of teachers; please correct me if I'm wrong.

The experimental results aren't convincing. There aren't any fair comparisons. For instance, in table 6 a WRN-28-10(sFEED) after 5 whole training iterations is compared to a WRN-28-1(BAN) after 1. It would be good to run BAN for as many iterations. A comparison to attention transfer (https://arxiv.org/abs/1612.03928) would be ideal for the ImageNet experiments. Furthermore, if one isn't interested in compression, then Table 4 indicates that an ensemble is largely preferable.

This work would benefit from a CIFAR-10 experiment as it's so widely used (interestingly, BANs perform poorly on CIFAR-10), also a task that isn't image classification would be helpful to get a feel of how the method generalizes.

In summary I believe this paper should be rejected, as the method isn't very novel, and the experimental merits are unclear.

Pros:
- Simple method
- Largely written with clarity

Cons:
- Method is not very novel
- No compared thoroughly enough to other work

---

> ### Author Response · Authors · 2018-11-26
> **Answer to reviewer #3**
>
> Thank you for the constructive review.
>
> First, the sFEED, though the purpose of BANs[1] and ours are different, sFEED in our work and BANs ended up with similar training structure, and we admit that it lacks its novelty for their similarity in architecture with BANs.
> However, for the pFEED, we did not, and could not use ‘ensemble of teachers’ that you noted, because our purpose is to deliver the knowledge of ensemble at feature map level without using teachers’ predictions.
>
> The purpose of Table 6 is not to show that our method beat others, but to show that ours are fairly good. We actually tried to reproduce the BANs, but unfortunately we could not reproduce the base DenseNet models that BANs used. They reported that error of base DenseNet-80-80 is 17.16, but our reproduction of it only could achieve 17.71. To the best of our knowledge, their DenseNet models such as DenseNet-80-80 or DenseNet-80-120 are not public release, and they are modulation of BANs, so we could not find differences in detail. The WRN-28-10 with first iteration at sFEED can be fair to compare with WRN-28-10 at BAN-1, but sFEED with 1 iteration is just FT without paraphraser.
>
> However, we think that the Table 4. can give fair and meaningful comparison. The KD column in Table 4 are those who really use ‘ensemble of teachers’ for training. Comparing ours with them shows that ours can be beneficial with giving specific knowledge with knowledges of ensemble at feature map level.
>
> For the CIFAR-10 experiment, I do not necessarily feels the importance to report since it already performs well on CIFAR-100, which is more difficult task, but I had experimented for few networks with sFEED, but did not report it on our paper. The result was
> ResNet-56   6.97 -> 6.16
> ResNet-110 6.43 -> 5.92
> WRN28-10   4.00 -> 3.62
>
> Testing on the tasks other than classification that you suggested would be good idea to try, and can be our future work.
>
> [1] Furlanello et al. Born-Again Neural Networks, 2018

---

> > ### Comment · AnonReviewer1 · 2018-11-29
> > **Reply to authors**
> >
> > Thank you for your reply.
> >
> > So as far as I can tell, pFeed is an ensemble of activations from teachers (or some function thereof) . This is effectively FitNets (https://arxiv.org/abs/1412.6550)/attention transfer with an ensemble of teachers.
> >
> > It is interesting to know that the BAN results aren't reproducible. I think when this happens, it's best to put your locally reproduced BAN results next to it and state that you weren't able to reproduce their results.
> >
> > Thank you for providing CIFAR-10 results.
> >
> > I will, however, stick with my original review score as I am still concerned about the lack of novelty of the methods.

---

### Official Review · AnonReviewer2 · 2018-10-31
**a good idea to try but the paper is not yet ready**

**Rating:** 4
**Confidence:** 3

**Review:**

In summary, I think this paper contains some reasonable results based on a reasonable, moderately novel, idea, but unfortunately, it is not yet ready for publication. Reading it made me rather confused.

Good things:
- The main idea is sensible, though distilling into the same architecture (sFEED) is not that novel. I think the pFEED is probably the more novel part.
- The numerical results are quite good.
- It's a fairly simple method. If others reproduced these results, I think it would be useful.

Problems:
- Some parts of the paper are written in a way that makes the reader confused about what this paper is about. For example the first paragraph. Some motivations I just did not understand.
- Some parts of the paper are repeating itself. For example "introduction" and "related works". The section on related work also includes some quite unrelated papers.
- The references in the paper are often pointing to work that came much later than the original idea or some pretty random recent papers. For example the idea of model compression (or knowledge distillation) is much older than Hinton et al. I believe it was first proposed by Bucila et al. [1] (which the authors mention later as if knowledge distillation and model compression were very different ideas), it definitely doesn't come from Kim et al. (2018). Learning from intermediate representations of the network is at least as old as Romero et al. [2]. Compression into a network of the same architecture is definitely older than Furnarello et al. (2018). It was done, for example, by Geras et al. [3]. The paper also cites Goodfellow et al. (2016) in some pretty random contexts. I don't want to be too petty about references, but unfortunately, this paper is just below a threshold that I would still find acceptable in this respect.
- The comparison in Table 6 would make more sense if the same architectures would be clearly compared. As it is, it is difficult to be certain where the improvement is coming from and how it actually compares to different methods.

Typos: Titap X, ResNext, prarphraser.

References:
[1] Bucila et al. Model Compression. 2006.
[2] Romero et al. FitNets: Hints for Thin Deep Nets. 2014.
[3] Geras et al. Blending LSTMs into CNNs. 2016.

---

> ### Author Response · Authors · 2018-11-26
> **Answer to reviewer #2**
>
> Thank you for your constructive feedback. I appreciate it and it was helpful in many points
>
> First, for the novelty issue, I admit that sFEED is not that novel because it is similar with BANs[1]. While we were working on our paper, we came across with BANs and found sFEED is similar with BANs, but I thought it is worth reporting it because sFEED performed better at ImageNet.
>
> Second, for the motivations, sorry for our bad at writing it clearly. We mentioned our point again at Answer to reviewer #1.
>
> For the related works and reference parts, we found that it was messy, and we will deliberately reflect what you pointed out.
>
> For the experiments, we agree on your point that it would be better if we had compared ours with other knowledge transfer algorithms that use information from feature map level(you pointed out AT[3])
> For the comparison on Table 6, since our purpose is to show that our method can deliver ensemble knowledge at feature map level, we just wanted to show that the performance is decent level.
> Actually, we tried, but failed to reproduce the base DenseNet[2] models that BANs[1] used for their base network which are their own modulation. Maybe the only one that can be compared is WRN-28-10 with BAN-1 and our sFEED with one iteration.
>
> Thanks for pointing out the typos.
>
> [1] Furlanello et al. Born-Again Neural Networks, 2018
> [2] Huang et al. Densely Connected Convolutional Networks, 2016
> [3] Zagoruyko et al. Paying More Attention to Attention: Improving the Performance of Convolutional Neural Networks via Attention Transfer, 2016

---

### Official Review · AnonReviewer3 · 2018-11-07
**Potentially lack of true novelty**

**Rating:** 5
**Confidence:** 3

**Review:**

I do not necessarily see something wrong with the paper, but I'm not convinced of the significance (or sufficient novelty) of the approach.

The way I understand it, a translator is added on top of the top layer of the student, which is nothing but a few conv layers that project the output to potentially the size of the teacher (by the way, why do you need both a paraphraser and translator, rather than making the translator always project to the size of the teacher which basically will do the same thing !? )
And then a distance is minimized between the translated value of the students and the teacher output layer. The distance is somewhat similar to L2 (though the norm is removed from the features -- which probably helps with learning in terms of gradient norm).

Comparing with normal distillation I'm not sure how significant the improvement is. And technically this is just a distance metric between the output of the student and teacher. Sure it is a more involved distance metric, however it is in the spirit of what the distillation work is all about and I do not see this as being fundamentally different, or at least not different enough for an ICLR paper.

Some of the choices seem arbitrary to me (e.g. using both translator and paraphraser). Does the translator need to be non-linear? Could it be linear? What is this mapping doing (e.g. when teacher and student have the same size) ? Is it just finding a rotation of the features? Is it doing something fundamentally more interesting?

Why this particular distance metric between the translated features? Why not just L2?

In the end I'm not sure the work as is, is ready for ICLR.

---

> ### Author Response · Authors · 2018-11-26
> **Answer to reviewer # 1.**
>
> Thanks for the review, but I think that your questions are out of focus. In the abstract of our paper, at the line 6, we noted that our algorithm is extension of FT[1], and we also mentioned in the third paragraph of the second page that we utilize the translator of FT.
> Most of the questions you ask are not obliged for us to answer them because our paper is not supposed to claim, but the FT paper could answer.
>
> Our point is:
> Using stronger teacher has many drawbacks. As Lan et al.(2018)[3] mentioned and we noted on our paper that, cases are possible where stronger teacher may not exist. Knowledge Transfer(KT) methods such as KD[2] can use ensemble teacher instead of stronger teacher with large numbers of parameters, but KT methods that does not use predictions(labels) are incompatible with label ensemble.
> What we claim is that our method could manage these problems, and also delivering knowledge at feature map layer may have advantages with giving more specific information, and could get decent results.
>
> [1] Kim et al. Paraphrasing Complex Network: Network Compression via Factor Transfer, 2018
> [2] Hinton et al. Distilling the Knowledge in a Neural Network, 2015
> [3] Lan et al. Knowledge distillation by on-the-fly native ensemble, 2018

---

### Meta-Review · Area_Chair1 · 2018-12-13
**lack of novelty**

**Confidence:** 5
**Recommendation:** Reject

**Metareview:**

The paper describes knowledge distillation methods. As noted by all reviewers, the methods are very similar to the prior art, so there is not enough novelty for the paper to be accepted. The reviewers' opinion didn't change after the rebuttal.